# Non-Responsive and Refractory Coeliac Disease: Experience from the NHS England National Centre

**DOI:** 10.3390/nu14132776

**Published:** 2022-07-05

**Authors:** Hugo A. Penny, Anupam Rej, Elisabeth M. R. Baggus, Sarah. H. Coleman, Rosalie Ward, Graeme Wild, Gerd Bouma, Nick Trott, John A. Snowden, Josh Wright, Simon S. Cross, Marios Hadjivassiliou, David S. Sanders

**Affiliations:** 1Academic Unit of Gastroenterology, Royal Hallamshire Hospital, Sheffield S10 2JF, UK; anupam.rej@nhs.net (A.R.); elisabeth.baggus@gmail.com (E.M.R.B.); shcoleman1@sheffield.ac.uk (S.H.C.); rosalie.ward@nhs.net (R.W.); graeme.wild1@nhs.net (G.W.); nick.trott@nhs.net (N.T.); simon.cross@nhs.net (S.S.C.); 2Department of Gastroenterology, Vrije Universiteit Medical Center, 1117 Amsterdam, The Netherlands; g.bouma@vumc.nl; 3Department of Haematology, Royal Hallamshire Hospital, Sheffield Teaching Hospital NHS Foundation Trust, Sheffield S10 2JF, UK; john.snowden1@nhs.net (J.A.S.); josh.wright5@nhs.net (J.W.); 4Department of Neurology, Royal Hallamshire Hospital, Sheffield Teaching Hospital NHS Foundation Trust, Sheffield S10 2JF, UK; m.hadjivassiliou@nhs.net

**Keywords:** non-responsive coeliac disease, refractory coeliac disease, gluten immunogenic peptides

## Abstract

We characterised the aetiology of non-responsive coeliac disease (NRCD) and provided contemporary mortality data in refractory coeliac disease (RCD) from our centre. We also measured urine gluten immunogenic peptides (GIPs) in patients with established RCD1 to evaluate gluten exposure in these individuals. Methods: This was a longitudinal cohort study conducted in Sheffield, UK. Between 1998 and 2019, we evaluated 285 adult (≥16 years) patients with NRCD or RCD. Patients with established RCD1 and persisting mucosal inflammation and/or ongoing symptoms provided three urine samples for GIP analysis. Results: The most common cause of NRCD across the cohort was gluten exposure (72/285; 25.3%). RCD accounted for 65/285 patients (22.8%), 54/65 patients (83.1%) had RCD1 and 11/65 patients (16.9%) had RCD2. The estimated 5-year survival was 90% for RCD1 and 58% for RCD2 (*p* = 0.016). A total of 36/54 (66.7%) patients with RCD1 underwent urinary GIP testing and 17/36 (47.2%) had at least one positive urinary GIP test. Conclusion: The contemporary mortality data in RCD2 remains poor; patients with suspected RCD2 should be referred to a recognised national centre for consideration of novel therapies. The high frequency of urinary GIP positivity suggests that gluten exposure may be common in RCD1; further studies with matched controls are warranted to assess this further.

## 1. Introduction

Coeliac disease (CD) is an immune-mediated enteropathy that develops in response to dietary gluten ingestion in genetically susceptible individuals [1]. Whilst most individuals will display improvement in signs and/or symptoms of CD after commencing a GFD, 7–30% of patients will continue to display clinical manifestations typical of CD and/or have persisting intestinal inflammation [2]. When this occurs for at least 6–12 months despite adherence to a GFD, these individuals are classified as having non-responsive CD (NRCD) [3].

NRCD covers a broad range of pathologies including ongoing gluten exposure or super-sensitivity to gluten [4,5,6]. Some individuals will present with apparent NRCD, but rather will have developed symptoms due to an associated pathology, such as microscopic colitis, small bowel bacterial overgrowth or irritable bowel syndrome (IBS) [6,7,8]. Alternatively, a subset of individuals will have developed a state of persistent mucosal inflammation despite strict adherence to a GFD and be classified as having refractory coeliac disease (RCD) [9].

RCD is categorised into two types: Type 1 (RCD1) and Type 2 (RCD2). RCD2 is typically identified based on the presence of an expanded population of “aberrant” small intestinal intra-epithelial lymphocytes (IELs) [10,11]. By contrast, there is no objective marker of RCD1, and the histological appearances of mucosal inflammation in RCD1 are similar to that in uncomplicated CD. Moreover, surrogate markers of gluten ingestion, such as dietary questionnaires and/or the presence of serological markers of CD (such as anti-tissue transglutaminase (tTG) and endomyseal antibodies (EMA)), do not accurately detect all dietary indiscretions [12,13]. Therefore, differentiating patients with true immune refractoriness to the GFD (i.e., RCD1) from those with persistent mucosal inflammation due to ongoing dietary gluten exposure is difficult, meaning the diagnosis and assessment of RCD1 is challenging.

Recent data suggest that testing for the presence of urinary gluten immunogenic peptides (GIPs), which comprise the immunodominant peptides in gluten, provides an accurate indication of gluten exposure over the preceding few days [14,15]. It has been proposed that measuring urine GIPs may help accurately identify true RCD1 [16]. However, this has only been evaluated in a recent small case series (*n* = 4) [16], and studies in larger cohorts of RCD1 are lacking.

The aims of this study were to (i) characterise the aetiology of NRCD in adult patients from the UK National Centre for NRCD and RCD; (ii) provide contemporary mortality data in RCD; and (iii) evaluate whether patients with established RCD1 have objective evidence of ongoing gluten ingestion, by evaluating the presence of urinary GIPs in these patients. In doing so, we aimed to provide clinically meaningful data to aid with the timely and accurate recognition and management of adults with CD and persisting symptoms.

## 2. Materials and Methods

### 2.1. Patient Inclusion

We first defined a cohort of patients with NRCD and RCD assessed at a single UK tertiary centre—Sheffield Teaching Hospitals NHS Foundation Trust, UK—which is an NHS England National Centre for NRCD and RCD. To achieve this, we performed prospective analysis of adult coeliac patients assessed for, and followed-up with, persisting signs and/or symptoms at the centre from 2008 to 2019. Retrospective analysis was undertaken of cases from 1998 to 2008. Thus, the study population comprised adult CD patients recruited over a 21-year period. The inclusion/exclusion criteria were as follows:

#### 2.1.1. Inclusion Criteria

(i)Age ≥16 years old.(ii)Confirmed diagnosis of CD: A diagnosis of typical CD was based on positive coeliac serology (tTG and/or EMA) and Marsh 3 histology on duodenal biopsy [17]. Seronegative CD was diagnosed in individuals with HLA-DQ2/DQ8-positivity, Marsh 3 histology on duodenal biopsies, negative coeliac serology, clinical and/or histological response to a GFD and no alternative causes for villous atrophy, as previously described [18].(iii)Patients assessed with persisting signs and/or symptoms: defined as failure of symptoms, signs or laboratory abnormalities typical of CD to improve despite at least 12 months of adherence to a GFD; or recurrence of symptoms, signs or laboratory abnormalities typical of CD despite initial response to a GFD [3].(iv)The identification of a pre-malignant/malignant lesion (e.g., UJ, RCD2 and EATL) at the time of index diagnosis of CD.

#### 2.1.2. Exclusion Criteria

(i)Patients who did not undergo repeat upper GI endoscopy with biopsies for investigation of persisting signs/symptoms.(ii)Incomplete or missing information regarding investigations of persisting signs/symptoms.

### 2.2. Patient Assessment and Diagnosis

All patients underwent repeat upper GI endoscopy with duodenal biopsies during their assessment. Further testing was performed at the discretion of the clinician and was based on the clinical presentation and/or findings from upper GI endoscopy. The diagnostic criteria of the different clinical entities are summarised in the Appendix A.

#### 2.2.1. Evaluation of the IEL Phenotype

Duodenal biopsies with evidence of persisting Marsh 3 lesions underwent detailed assessment of the IEL phenotype. This was determined by immunohistochemistry (IHC) assessment of IELs on formalin-fixed paraffin-embedded duodenal tissue sections and/or flow cytometry evaluation of small intestinal IELs. Flow cytometry has been incorporated into routine clinical use at our centre since 2018.

#### 2.2.2. IHC

IHC was performed on sections from paraffin-embedded biopsy specimens using antibodies directed against CD3 (Clone: LN10; Leica Concentrate), CD8 (Clone: C8/44B; Dako RTU Link) and CD4 (Clone: 4B12; Dako RTU Link). RCD2 was diagnosed based on the presence of ≥40% aberrant IELs (CD3+ CD8− cells) as previously described (Appendix A) [11].

#### 2.2.3. Flow Cytometry

IELs were isolated and prepared from duodenal biopsies acquired during endoscopy as previously described [11,19]. Multicolour staining of lymphocytes was performed using APC-Cy7-labelled anti-CD45 (clone: 2D1; BD Bioscience, EU), PE-labelled anti-CD7 (clone: M-T701; BD Bioscience, EU), FITC-labelled anti-CD103 (clone: Ber-ACT8; BD Bioscience, EU), PerCP-Cy5.5-labelled anti-surface(s)CD3 (clone SK7; BD Bioscience, EU) and PE-Cy7-labelled anti-cytoplasmic(c)CD3 (clone: SK7; BD Bioscience, EU). Flow cytometric analysis was performed using a Fortessa flow cytometer (BD Bioscience, Oxford, UK). RCD2 was diagnosed based on the presence of ≥20% aberrant IELs (CD45+ CD7+ sCD3− cCD3+ cells) as previously described (Appendix A) [11].

#### 2.2.4. Clonality Studies

Duodenal samples that were identified as having an aberrant IEL phenotype meeting the RCD2 criteria were sent offsite for molecular detection of clonal TCR chain rearrangements, using DNA extracted from intestinal biopsy specimens, by multiplex polymerase chain reaction, as previously described [20].

### 2.3. Evaluation of Urinary GIPs in Patients with RCD1

Individuals with established RCD1 who had persisting mucosal inflammation on duodenal biopsies from their most recent upper GI endoscopy and/or ongoing symptoms at clinical review were invited to provide three individual urine samples (one during the week and two on the weekend) between November 2019 and December 2019. Providing multiple samples over different days enabled us to account for potential changes in dietary habit/gluten consumption during the week and on weekends [14,15]. Patients were aware that they were providing their urine samples for ongoing investigation and monitoring of RCD.

Urine samples were transported to the lab on the same day of collection. In the lab, urine samples were processed for testing using the iVYDAL GIP urine immunochromatographic point of care test (PoCT) (Biomedal; Seville, Spain), according to the manufacturer’s instructions. In brief, 70 μL of urine was mixed with 30 μL of conditioning solution and the 100 μL volume was added to the sample zone on the test cassette; cassettes were then read manually according to the manufacturer’s instructions. If present, GIPs in the sample interact with monoclonal antibodies A1 and G12 on the test strip, producing a red line in the test zone [15]. The manufacturer has previously reported sufficient GIP stability in urine for at least 24 h at room temperature. The limits of GIP quantification in urine using this test have been previously discussed [15].

### 2.4. Ethical Considerations

Patients underwent clinical tests and assessments as part of their routine care. The data collection was approved by the Yorkshire and the Humber Sheffield Research Ethics Committee, under registration number 14/YH/1216.

### 2.5. Statistical Analysis

Data handling was conducted using spreadsheets in Microsoft Excel. Data are presented as mean plus/minus standard deviation or median with the interquartile range (IQR), or numeral range, as indicated. Statistical analysis and graph construction were performed using GraphPad Prism version 7 (GraphPad Software, Inc., San Diego, CA, USA). A *p* value of <0.05 was considered statistically significant.

## 3. Results

### 3.1. Ongoing Dietary Gluten Exposure Is the Most Common Cause of NRCD

A total of 2553 adult patients with suspected or confirmed CD were seen at our centre during the study period. In total, 320/2553 patients (12.5%) had persisting symptoms; of these, 7 patients (2.2%) were found to have an incorrect index diagnosis of CD, 10 patients (3.1%) did not undergo repeat gastroscopy and/or further investigations and 18 patients (5.6%) had missing information, so were excluded from further analysis. Therefore, the final cohort for analysis comprised 285 patients (Figure 1A); 69/285 patients (24.2%) were out-of-region referrals into our centre (Figure 1B).

Of the 285 patients, 198 (69.5%) were female, and the median age at index CD diagnosis was 42 years (IQR 26–53 years). A list of the aetiologies of NRCD is presented in Table 1. Ongoing dietary gluten exposure was the most common cause of NRCD (72/285 patients; 25.3%). IBS/functional causes were the next most frequent cause of persisting symptoms across the cohort (64/285 patients; 22.5%). Interestingly, conditions associated with CD accounted for persisting symptoms in half (143/285 patients; 50.2%) of the cohort. In total, 65/285 patients (22.8%) were diagnosed with RCD; 44/65 patients (67.7%) with RCD were out-of-region referrals. RCD1 accounted for 54/65 patients (83.1% of RCD cases) and 11/65 patients (16.9%) had RCD2.

### 3.2. RCD2 and EATL Are Associated with Greater Mortality than RCD1

Of patients with RCD, a greater proportion of women had RCD1 (40/65 (61.5%) female vs. 14/65 (21.5% male), while a greater proportion of men had RCD2 (4/65 (6.2%) female versus 7/65 (10.8%) male; *p* = 0.0372). The median age of patients with RCD2 was greater both at index diagnosis of CD and at diagnosis of RCD (56 years and 62 years old, respectively) than those with RCD1 (44 years and 53 years old, respectively; *p* < 0.05) (Figure 2A). Patients with RCD2 had a shorter time interval between index diagnosis of CD and diagnosis of RCD than those with RCD1 (median 2-year interval versus 5 years, respectively; *p* < 0.05) (Figure 2B).

Patients with RCD1 were most commonly treated with either open capsule budesonide or thiopurines (Table 2). Those with RCD2 were treated with open capsule budesonide, thiopurines, mycophenolate or cladribine. Some patients were undergoing further evaluation off therapy at the time of the study (Table 2) but had previously been treated with steroids and/or thiopurines.

Patients with RCD2 were more likely to die within the study period compared to patients with RCD1 (all-cause mortality: 5/54 patients (9.6%) with RCD1 versus 4/11 patients (36.4%) with RCD2; *p* = 0.037). The estimated 5-year survival was 90% in RCD1 and 58% in RCD2 (Figure 3A).

One patient with RCD2 developed secondary EATL during follow-up, 15 months after the diagnosis of RCD2 was made. Three patients were diagnosed with de novo EATL. All patients with de novo EATL were male and had a median age at index CD diagnosis of 46 years (range 3–57 years). Median time from index diagnosis of CD to diagnosis of de novo EATL was 16 years (range 2–66 years). The estimated 5-year survival for all patients with EATL (one secondary EATL; three de novo EATL) was 50.0% (Figure 3B). Five-year survival increased to 66.7% when only patients with de novo EATL were considered (data not shown).

### 3.3. Evidence of Ongoing Gluten Exposure in Individuals with RCD1

Forty-four patients with RCD1 had evidence of ongoing mucosal inflammation on their latest biopsies and/or ongoing symptoms; eight patients were excluded from further analysis due to patient choice and/or incomplete testing. The remaining 36 patients provided three urine samples that were subjected to GIP testing (Figure 4A,B); clinical and demographic information are listed in Table 3.

Of 36 patients, 17 (47.2%) had evidence of at least one positive test (Figure 5A). An analysis of the number of positive tests for each individual revealed that six patients had evidence of GIPs in all three independent urine tests, representing 16.7% of the total cohort (Figure 5B). Furthermore, we found no difference in the frequency of GIP positivity during the week compared with the weekend (data not shown).

## 4. Discussion

In this study, we found that the most common cause of persisting symptoms in the largest cohort of adult NRCD to date is ongoing dietary gluten exposure and that the contemporary mortality data in RCD2 remains poor. We also identified a high frequency of positive urinary GIP tests, which suggests that ongoing gluten exposure may be common in patients with established RCD1. However, further studies with appropriately matched controls are warranted to investigate this finding further.

Prior studies have shown that dietary gluten is the most common cause of NRCD and occurs in 22% to 51% of cases [4,5,6,7]. Our data confirms the validity of these earlier observations and, in doing so, highlights that maintaining a strict GFD remains a challenge for some patients, despite more recent widespread availability of GF products both in stores and online [22]. Close dietetic and psychological counselling, as well as online education modules have been proposed as ways to enhance GFD adherence [23]. These should be considered for patients with CD and/or a history of poor GFD adherence. Notably, our data also suggest that CD-associated conditions accounted for half of all cases referred into the clinic with persisting symptoms. This finding highlights the importance of adopting a systematic approach when considering the cause of persisting symptoms in CD, and diagnostic approaches have recently been outlined [3].

We found that 5% of the cohort had RCD2 and/or de novo EATL. It is noteworthy that this does not represent the true prevalence of these conditions in NRCD. We attribute this to referral bias due to our centre being a recognised referral site for RCD. The data in RCD and EATL are limited, as these are rare conditions with an estimated annual incidence in Western Europe of 0.031/100,000 and 0.1/100,000, respectively [6]. Our findings, that patients with RCD2 are more likely to be older men diagnosed with CD in later life, aligns with previous findings [20,24,25,26] and may help risk-stratify patients seen in clinic with persisting symptoms. Prognosis in RCD2 is reportedly poor, with studies from major coeliac centres worldwide reporting 5-year survival rates of 58% (Amsterdam, The Netherlands) [20], 44% (Paris, France) [24], 45% (Rochester, NY, USA) [25] and 53% (Berlin, Germany) [26]. Our contemporary data in RCD2 from the UK align with these earlier findings. The similarity between the mortality data in RCD2 presently and data of studies conducted more than a decade ago may reflect that the treatment of RCD2 has not evolved over this time-period and that new approaches are urgently needed. Autologous haemopoietic stem cell transplantation (aHSCT), now indicated in a range of autoimmune and inflammatory diseases [21], has been used to treat RCD2 [27,28], with limited evidence that it may prevent progression to EATL. No patients in the present study received aHSCT, and this approach is not currently funded by the NHS in RCD2. However, the small number of cases and poor prognosis of RCD2 warrants patient referral to national specialist centres that have appropriate infrastructure for consideration of novel therapies, including aHSCT, to improve outcomes.

To date, the diagnosis of RCD1 typically relies upon careful dietary review and excluding elevated circulating tTG/EMA titres [10]. However, clinical assessment of dietary habits can be inaccurate at detecting ongoing dietary gluten exposure [29,30]. Furthermore, coeliac serology lacks sensitivity in the follow-up of CD [12,13], and mucosal healing can take years to achieve [4,5], meaning it is difficult to differentiate ongoing gluten exposure from slow healing, from RCD1 in some patients. Several recent studies have documented the utility of measuring urine and/or faecal GIPs as a marker of dietary gluten adherence in uncomplicated CD [14,15,16,29,30,31,32]. These studies have reported a window of detection for urine tests ranging from 4 h to 48 h after gluten ingestion [14,29], and a sensitivity of 95% following gluten doses of 2 g or more [33].

Interestingly, we found that a high proportion of the RCD1 cohort had a positive urine GIP test. In a similar manner, Moreno et al. recently showed that three of four patients diagnosed with RCD1 who had persisting duodenal inflammation and CD symptoms had GIPs in their urine, which was suggestive of ongoing gluten exposure [16]. In accordance with previous studies, three urinary samples were collected presently for GIP testing as it has been shown that this may increase the sensitivity of the tests when detecting gluten ingestion [33]. Furthermore, we included weekday and weekend sampling to account for different dietary habits and/or gluten vigilance that may occur at different times over the course of a week [29].

On the one hand, these findings could indicate persisting inadvertent gluten exposure that was not identified during previous assessments, which may confer doubt over the index RCD1 diagnosis. On the other, they may indicate dietary lapses in patients with established RCD1. It is noteworthy, however, that we did not assess symptoms or duodenal inflammation at the time of GIP testing in the present study. Therefore, it is not clear whether GIP positivity translates into a clinically meaningful result. Indeed, recent findings suggest that trace amounts of gluten that are tolerable to some and fall within the recommendations of the GFD may cause a positive result on urine GIP tests [34]. Furthermore, others have found a high false-negative rate of GIP tests, which may relate to individual differences in gluten metabolism [35]. Therefore, further studies with appropriately matched controls are required to evaluate the accuracy of GIP tests in RCD1. This is important to understand, as mislabelling RCD1 in the setting of ongoing gluten exposure has possible implications for inappropriate administration of medications, may heighten patient anxiety, and may increase the cost and caseload of monitoring these individuals [16]. Moreover, the differences in mortality between RCD subtypes may in part be because of mislabelling individuals as RCD1. Thus, further work focusing on improving diagnostic accuracy is important to ascertain the true morbidity and mortality associated with RCD1.

Notably, as this is a single-centre study, the results may be subject to selection bias. Moreover, as the study comprised a 21-year period, there was heterogeneity in the diagnostic assessment and follow-up of patients.

The growing number of patients being diagnosed with CD means that NRCD and RCD are likely to be an increasing problem. We advocate the management of individuals with suspected RCD2 at recognised national centres for consideration of novel therapies. Further studies with appropriately matched controls are warranted to assess whether urinary GIP testing may have a role in the assessment of RCD1. This is important for the accurate diagnosis and monitoring of this condition.

## 5. Conclusions

With the growing number of patients being diagnosed with CD, NRCD and RCD are likely to be an increasing problem. The poor prognosis of RCD2 warrants management of individuals at recognized national centres for consideration of novel therapies, including aHSCT, to improve outcomes. Further studies with appropriately matched controls are warranted to assess whether urinary GIP testing may have a role in the assessment of RCD1. This is important for the accurate diagnosis and monitoring of this condition.

## Figures and Tables

**Figure 1 nutrients-14-02776-f001:**
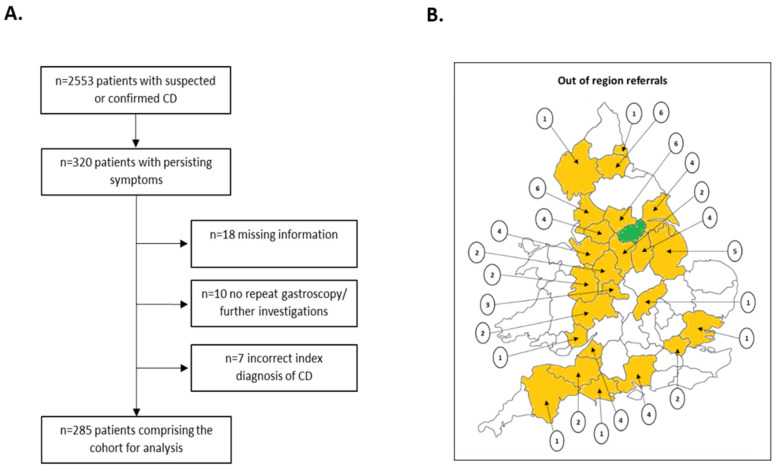
(**A**) Flow chart detailing the study patient selection. (**B**) Map of England illustrating the frequency and location of the 69 out-of-region referrals. The counties patients were referred from are highlighted in orange; the encircled numbers represent the number of patients referred from each country; South Yorkshire/Sheffield is highlighted in green. CD = coeliac disease.

**Figure 2 nutrients-14-02776-f002:**
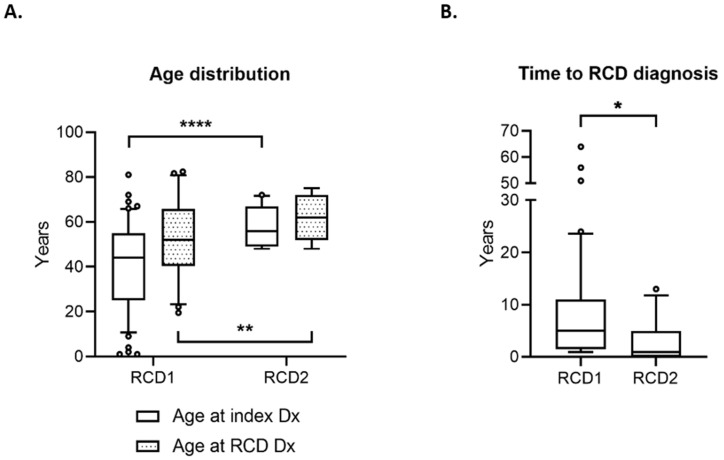
Comparison of patient demographics in RCD. (**A**) Box and whisker plot of age distribution of patients with RCD1 and RCD2; error bars indicate 10th–90th percentile; outliers are plotted as individual data points; comparisons between groups were conducted using two-way ANOVA with post hoc Tukey test. (**B**) Box and whisker plot of time to RCD diagnosis from index CD diagnosis; error bars indicate 10th–90th percentile; outliers are plotted as individual data points; comparison between groups was conducted using the Mann–Whitney test. * = *p* < 0.05; ** = *p* < 0.01; **** = *p* < 0.0001. Dx = diagnosis.

**Figure 3 nutrients-14-02776-f003:**
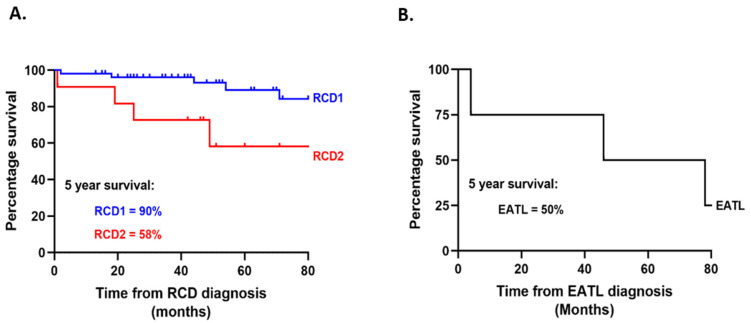
(**A**) Survival analysis in RCD1 and RCD2. The estimated 5-year survival in RCD1 was 90% and RCD2 was 58% (*p* = 0.016); comparison was analysed using the log–rank test (Mantel–Cox method). (**B**) Survival analysis in all patients with EATL (one secondary EATL; three de novo EATL). The estimated 5-year survival in EATL was 50%.

**Figure 4 nutrients-14-02776-f004:**
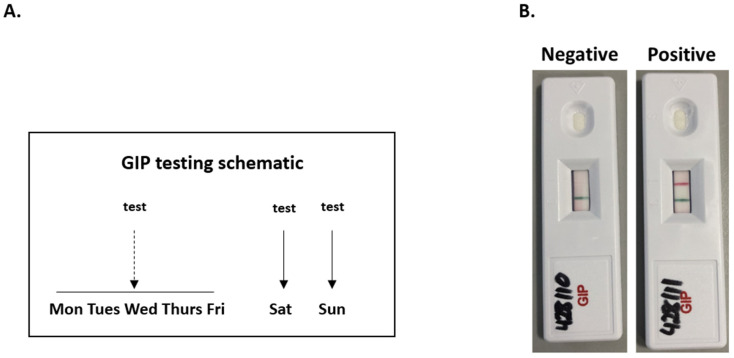
(**A**) Urine GIP testing schematic. Patients provided three urine samples in total—one from during the week and two from the weekend. (**B**) Example urine GIP immunochromatographic tests. If present, GIPs in the sample interact with monoclonal antibodies A1 and G12 on the test strip, producing a red line in the test zone [21]. An internal control generates a green line which indicates correct test performance [21].

**Figure 5 nutrients-14-02776-f005:**
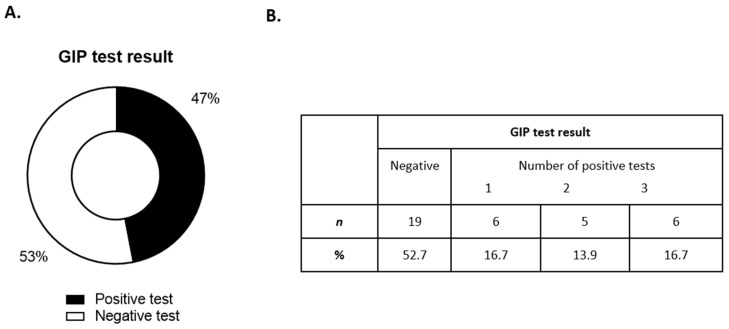
(**A**) Frequency of patients with a positive or negative urine GIP test result. (**B**) Frequencies of number of positive tests.

**Table 1 nutrients-14-02776-t001:** The causes of NRCD in patients with established CD.

Aetiology	Frequency N (% of Cohort)
Dietary indiscretion	72 (25.3%)
Supersensitive	6 (2.1%)
RCD1	54 (18.9%)
RCD2	11 (3.9%)
De novo EATL	3 (1.1%)
Small bowel DLBCL	2 (0.7%)
Small bowel adenocarcinoma	2 (0.7%)
GORD	4 (1.4%)
H pylori gastritis	4 (1.4%)
Lactose/fructose intolerance	8 (2.7%)
SBBO	7 (2.5%)
Reflux dysmotility	13 (4.6%)
IBS/functional	64 (22.5%)
Pancreatic exocrine insufficiency	8 (2.7%)
Bile acid diarrhoea	2 (0.7%)
Microscopic colitis	12 (4.2%)
Inflammatory bowel disease	3 (1.1%)
Other *	10 (3.5%)

* Other = one each of thyroid disease, allergy, liver haemangioma, diverticulitis, small intestinal erosions, small bowel neuroendocrine cancer, oesophageal adenocarcinoma, psychiatric, unclassified. DLBCL = diffuse large B cell lymphoma; GORD = gastro-oesophageal reflux disease; SBBO = small bowel bacterial overgrowth; IBS = irritable bowel syndrome.

**Table 2 nutrients-14-02776-t002:** Current treatment in patients with RCD1 and RCD2.

	RCD 1 N (%)	RCD 2 N (%)
Open capsule budesonide	16 (29.6%)	2 (18.2%)
Azathioprine/6-MP	23 (42.6%)	3 (27.2%)
Mycophenolate	0 (0%)	2 (18.2%)
Cladribine	0 (0%)	2 (18.2%)
None at present	15 (27.8%)	2 (18.2%)

**Table 3 nutrients-14-02776-t003:** Clinical and demographic details of patients diagnosed with RCD1 who underwent urinary GIP testing.

	Patient Cohort
Mean age, years (SD)	57.5 (15)
Female:male	25:11
Main symptoms at latest clinic review *	% (*n*)
Gastrointestinal	29.7% (11)
Extra-intestinal	13.5% (5)
Malabsorptive	21.6% (8)
Labs at latest clinic review	% (*n*)
Anaemia	16.2% (6)
Iron deficient	18.9% (7)
Marsh score on most recent follow-up biopsy	% (*n*)
0	5.5% (2)
1&2	2.8% (1)
3	91.7% (33)

* Gastrointestinal symptoms include abdominal pain, dyspepsia, dysphagia, nausea, vomiting, bloating and constipation; extra-intestinal symptoms include ataxia, headaches, dizziness, fatigue, joint pain and skin rash; malabsorptive symptoms include diarrhoea, haematinic deficiencies with/without anaemia and weight loss.

## Data Availability

Data availability considered upon specific request.

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
