# Peer review of "Non-Responsive and Refractory Coeliac Disease: Experience from the NHS England National Centre"

_nutrients, 2022, doi:10.3390/nu14132776_

Round 1
Reviewer 1 Report
This was an interesting study concerning non-responsive and refractory coeliac disease. The paper is well-written, study setting seems appropriate and the cohort size adequate. Few minor comments:
As you say, NRCD covers a broad range of pathologies and there is no marker that could objectively define patient with RCD1. Please describe, how they were defined in this study. I mean, were they characterized as having RCD1, if no other reason could be identified behind the ongoing symptoms or persisting mucosal damage?
Also, is it possible that there is such a big difference in the survival between different RCD types partly because some of the patients characterized as having RCD1 were not actually suffering from RCD but were patients with ongoing gluten consumption?
Please consider also making the paper easier to read by minimazing page breaks in the middle of the tables, and by placing the figures on the same page as the figure legend. There is also a problem in numbering of the criteria in section 2.1, there are hyphens in the middle of the words and in the abstract all of the subheadings are not all bolded.
Author Response
As you say, NRCD covers a broad range of pathologies and there is no marker that could objectively define patient with RCD1. Please describe, how they were defined in this study. I mean, were they characterized as having RCD1, if no other reason could be identified behind the ongoing symptoms or persisting mucosal damage?
Many thanks for raising this point. As the reviewer highlights, there is no objective marker for RCD1, rather it is currently accepted that patients are labelled as RCD1 if they meet a set number of criteria, which include:
Typical clinical manifestations of CD plus evidence of Marsh 3 lesions on repeat duodenal biopsy, that have persisted, or recurred, despite strict adherence to a GFD for at least 12 months; the IEL phenotype within the duodenal tissue is within normal limits; and other causes of villous atrophy have been excluded.
We have now included this explanation, along with other diagnostic criteria used within the current study, in Supplementary data - Table 1.
Also, is it possible that there is such a big difference in the survival between different RCD types partly because some of the patients characterized as having RCD1 were not actually suffering from RCD but were patients with ongoing gluten consumption?
This is a really interesting point. Given the difficulty in diagnosing RCD1 as discussed, it is entirely feasible that some patients with RCD1 may indeed actually have ongoing gluten consumption, and our GIP data is in support of this. It would then follow that outcome data in RCD1 does not reflect true morbidity and mortality in this subtype.
We have included the following text in the discussion to illustrate this point;
Page 10, Line 373:
"...the differences in mortality between RCD subtypes may in part be because of mislabeling individuals as RCD1. Thus, further work focusing on improving diagnostic accuracy is important to ascertain the true morbidity and mortality associated with RCD1."
However, it is notable that our data on these differences does align with previous findings from other coeliac centres across Europe and North America, and so is reflective of current understanding.
Please consider also making the paper easier to read by minimazing page breaks in the middle of the tables, and by placing the figures on the same page as the figure legend. There is also a problem in numbering of the criteria in section 2.1, there are hyphens in the middle of the words and in the abstract all of the subheadings are not all bolded.
Many thanks for pointing this out, we have re-formatted to try and make it easier to read as suggested, while maintaining the flow of the paper. We have also amended the section numbering to align with the journal requirements.
Reviewer 2 Report
I see some points needing the authors attention:
Materials and methods
Inclusion citeria: As there are some uncertainties with seronegative CD diagnosis, I miss some data about the number of seronegative patients and a more detailed explanation of how these patients were diagnosed (IHQ, cytometry, histologic response to gluten). Risk factor means genetics?
I am not sure inclusión criterion iv is really an inclusión crterion.
In line 102 the authors mention diagnostic criteria in Suppl data table 1 but I cannot find them.
Probably It is the hardest point. Near 20% of patients with NRCD had a RCD-1. RCD-2 has objective markers but RCD-1 does not. Considering the sometimes inespecific symptoms of patients and the persistence of villous atrophy even longer tan 2 years, the author should show the RCD-1 diagnostic criteria used (44 patients). Just symptoms and inflammation in the biopsies? Malabsorption data? Weight loss? Atrophy persistence longer tan a mínimum of time?
Results.
Line 177. What do the authors mean with: “conditions associated with celiac disease”? (also mentioned in line 262).
Line 179. What does the authors mean saying: “ 44/65 (67,7%) patients with RCD were out of region referrals”? Are there data missing?
Table 2.
Please pay attention to the interlinear spaces in right column in Table 2.
Three RCD patients did not have Marsh 3. Did you make a RCD diagnosis without villous atrophy? Authors should explain this.
Discussion.
Please pay attention to line 268-270. Are correct the rates showed? It seems to be more frequent lymphoma tan RCD. Is missing a “2”, ¿”RCD-2”?
Some information about RCD-1 and RCD2 treatments experienced by the authors should be summarized.
Author Response
Inclusion citeria: As there are some uncertainties with seronegative CD diagnosis, I miss some data about the number of seronegative patients and a more detailed explanation of how these patients were diagnosed (IHQ, cytometry, histologic response to gluten). Risk factor means genetics?
Many thanks for raising this point; we have now amended the text to illustrate/ explain fully how individuals with seronegative coeliac disease were diagnosed.
Methods; Page 2; Line 83:
"Seronegative CD was diagnosed in individuals with HLA-DQ2/DQ8-positivity, Marsh 3 histology on duodenal biopsies, negative coeliac serology, clinical and/ or histological response to a GFD, and no alternative causes for villous atrophy, as previously described [18]."
I am not sure inclusión criterion iv is really an inclusión crterion.
We included this to reflect that not only individuals with persisting symptoms/ signs (ie 12 months or more post-index CD diagnosis) were included in the study, but also those who were diagnosed with pre-malignant/ malignant lesions identified at presentation/ index diagnosis of coeliac disease. Please let us know if you would like further clarification about this and we can ammend as appropriate.
In line 102 the authors mention diagnostic criteria in Suppl data table 1 but I cannot find them.
Many thanks for highlighting this; There is a supplementary file that was uploaded, with diagnostic criteria used presently, as well as flow cytometry and histology examples. We have made sure this is uploaded with the revision and that the journal/ website has the file available.
Probably It is the hardest point. Near 20% of patients with NRCD had a RCD-1. RCD-2 has objective markers but RCD-1 does not. Considering the sometimes inespecific symptoms of patients and the persistence of villous atrophy even longer tan 2 years, the author should show the RCD-1 diagnostic criteria used (44 patients). Just symptoms and inflammation in the biopsies? Malabsorption data? Weight loss? Atrophy persistence longer tan a mínimum of time?
Thank you for asking us to clarify this point further. As the reviewer correctly points out, and the paper itself highlights, the diagnosis of RCD1 is difficult as there is no objective marker that differentiates RCD1 from slow mucosal healing, or even mucosal inflammation associated with inadvertent on-going gluten ingestion. We follow the currently accepted methods for diagnosing RCD1 and have included an explanation of this in Supplementary Table 1, as below:
Supplementary Table 1:
"RCD1 diagnosis - Typical clinical manifestations of CD plus evidence of Marsh 3 lesions on repeat duodenal biopsy, that have persisted, or recurred, despite strict adherence to a GFD (as assessed by a specialist dietitian) for at least 12 months. The IEL phenotype within the duodenal tissue is within normal limits. Other causes of villous atrophy have been excluded.[4,10]".
Using the above criteria, the typical manifestations of coeliac disease may indeed include, but are not limited to, weight loss and/ or biochemical evidence of malabsorption. As it can be unclear, we rely on experience to interpret this clinical scenario and the present criteria allows for this. We hope this explains more fully the situation, although agree that further work is needed to help improve the accuracy of the diagnosis of RCD1.
Line 177. What do the authors mean with: “conditions associated with celiac disease”? (also mentioned in line 262).
As the reviewer is likely aware, a number of conditions (such as microscopic colitis, IBS, intolerances, etc) are more common in patients with coeliac disease than an otherwise healthy population. We were trying to highlight this point in the manuscript. If the reviewer feels this is ambiguous or incorrect, we would be happy to consider revising/ editing these sentences.
Line 179. What does the authors mean saying: “ 44/65 (67,7%) patients with RCD were out of region referrals”? Are there data missing?
This was included to highlight that we are a national referral site for coeliac disease, and often get referrals from across the UK of patients with suspected complex CD/ RCD. It also illustrates, as is highlighted in the discussion, that there is a selection bias for RCD meaning one cannot interpret prevalence data for RCD from this cohort.
Please pay attention to the interlinear spaces in right column in Table 2.
Thank you for highlighting this - we have now amended this Table.
Three RCD patients did not have Marsh 3. Did you make a RCD diagnosis without villous atrophy? Authors should explain this.
Thank you for asking us to clarify this point. Table 2 are data relating to latest clinic review/ follow-up biopsy of patients with an established diagnosis of RCD. They were included in the GIP aspect of the study if they had ongoing villous atrophy and/ or had ongoing symptoms at latest clinical review. Therefore, these 3 individuals with established RCD1 had ongoing symptoms, but not demonstrable evidence of atrophy on latest biopsy. We have included an explanation of this in the text as follows:
Results; Page 7; Line 247:
"Forty-four patients with RCD1 had evidence of ongoing mucosal inflammation on their latest biopsies and/ or on-going symptoms..."
Please pay attention to line 268-270. Are correct the rates showed? It seems to be more frequent lymphoma tan RCD. Is missing a “2”, ¿”RCD-2”?
Thank you for highlighting this. Four patients had lymphoma in the present study, one of which had a prior diagnosis of RCD2, but three of which were de novo EATL, which is why it may appear that way. We have reflected this for clarity both in the text and Table 1.
Some information about RCD-1 and RCD2 treatments experienced by the authors should be summarized.
This is indeed an important point. We have now summarised the treatment in both the text as outlined below, and in Table 2.
Results; Page 6; Line 214:
"Patients with RCD1 were most commonly treated with either open capsule budesonide, or thiopurines (Table 2). Those with RCD2 were treated with open capsule budesonide, thiopurines, mycophenolate, or cladribine. A proportion of patents were undergoing further evaluation off therapy at the time of the study (Table 2), but had previously been treated with steroids and/ or thiopurines."
Reviewer 3 Report
I am very happy to read a paper of one of the most important center for Celiac Disease in the world. I ask to the authors more details about the therapy in Refractory Type 2 Celiac Disease and Intestinal Lymphoma
Author Response
I am very happy to read a paper of one of the most important center for Celiac Disease in the world. I ask to the authors more details about the therapy in Refractory Type 2 Celiac Disease and Intestinal Lymphoma
Many thanks to the reviewer for their kind comments about the paper. We also thank them for highlighting this important point and have included details about the treatment of patients with RCD1 and RCD2 in the text as outlined below, and have also summarized this information in Table 2.
Results; Page 6; Line 214:
"Patients with RCD1 were most commonly treated with either open capsule budesonide, or thiopurines (Table 2). Those with RCD2 were treated with open capsule budesonide, thiopurines, mycophenolate, or cladribine. A proportion of patents were undergoing further evaluation off therapy at the time of the study (Table 2), but had previously been treated with steroids and/ or thiopurines."
We have only provided broad details about treatment in the present paper, as this was meant as an observational study and was conducted over a long time period. As the data illustrates, we only had a very small cohort of patients with EATL. Of these, one patient underwent a laparotomy with post-operative chemotherapy; one patient had CHOP chemotherapy (cyclophosphamide, doxorubicin, vincristine and prednisone) and one patient underwent GEMOX chemotherapy (gemcitabine and oxaliplatin), followed by IVE/ MTX treatment (ifosfamide, etoposide, epirubicin/ methotrexate). As these individuals underwent quite different individualised treatment regimes, and it is such as small cohort, we felt including this data didnt add to the current study. Rather, RCD/ EATL treatment is the focus of ongoing work at our Centre.
We hope that this is acceptable to the reviewer; if they feel that this information would add to the manuscript, then please let us know and we will revise the manuscript as appropriate.